# Recent Reminders with Word-Image Information Can Improve Children’s Prospective Memory Performance

**DOI:** 10.3390/bs15091258

**Published:** 2025-09-15

**Authors:** Yan Yang, Yunfei Guo, Mingyuan Wang

**Affiliations:** 1School of Art, Southeast University, Nanjing 211189, China; 230179677@seu.edu.cn; 2Faculty of Education, Henan University, Kaifeng 475001, China; gyf@henu.edu.cn

**Keywords:** prospective memory, children, reminder, attention load, cue monitoring

## Abstract

Children’s prospective memory is not yet mature, and setting reminders is an effective method to improve their prospective memory. This study aimed to explore how reminders, placed at different distances from prospective memory cues, affect children’s prospective memory under different attention load conditions. A total of 170 primary school students aged 7–12 (*M* = 9.54, *SD* = 1.68) took part in the experiment in a laboratory environment. The experimental program was presented using E-prime 2.0 on one desktop computer. This study used a 3 (reminder conditions: recent reminder, distant reminder, control condition) × 2 (attention loads: low, high) between-subjects design. The results showed that in both low and high attention load conditions, the accuracy of prospective memory in the recent reminder condition was much higher than that in both the distant reminder and control conditions. The accuracy of ongoing tasks under the recent reminder condition was also significantly higher than that under the distant reminder and control conditions. The results showed that recent reminders can improve children’s prospective memory performance while reducing attentional expenditure, and the promoting effect of recent reminders on children’s prospective memory was not affected by attentional loads.

## 1. Introduction

Prospective memory is defined as the capacity to remember to execute planned actions in the right situations ([7]; [10]). For instance, remembering to submit homework when you see your teacher. The preparatory attention processing and memory processing (PAM) theory suggests that prospective memory involves two types of processing processes: preparatory attention processing and memory processing. Among them, preparatory attention processing involves top-down attention monitoring, while memory processing involves bottom-up spontaneous retrieval ([20]). Prospective memory comprises two processing components: a prospective component and a retrospective component. According to the PAM theory, the prospective component involves cue monitoring, while the retrospective component involves memory processing ([11]). Successful cue monitoring and intention retrieval are both indispensable for the successful execution of prospective memory tasks. Prospective memory undergoes rapid development during the school-age period, but it remains immature in early primary school children ([28]). Prospective memory tasks frequently appear in children’s daily routines, and children’s performance on these tasks can affect their quality of life. However, studies from multiple countries around the world have found that the prospective memory ability of school-age children is significantly lower than that of adults ([6]; [29]). The prospective memory performance of Chinese primary school children also shows significant deficiencies ([28]). Therefore, to support reliable performance of prospective memory in children, some effective strategies can be implemented. Among these, reminders constitute a potent intervention.

Existing studies generally find that reminders can enhance children’s prospective memory ([4]; [5]; [16]). However, the reminders employed in these studies have invariably been presented in the form of written language or pictures. In addition to the reminder of the content, the reminder of the context in which the cues of prospective memory appear may also significantly improve children’s performance in prospective memory. The effects of content and situational reminders on prospective memory are not identical. For example, writing the details of the intended action on a sticky note constitutes a content reminder, which can provide help when an individual forgets the content of the prospective memory task. Setting a prominent alarm before the appearance of prospective memory cues is a reminder of the situation, which can help individuals know when prospective memory cues appear. According to the PAM theory, situational reminders can significantly reduce the dependence of cue monitoring on attention. The development of the prefrontal cortex in children is not yet fully mature ([8]), and they have difficulty in flexibly allocating their attention among multiple tasks ([27]). Successfully carrying out prospective memory tasks usually demands considerable attentional resources, especially during cue monitoring ([24]). When attentional resources are insufficient, children may have a hard time giving enough attention to cue monitoring and therefore have difficulty completing prospective memory tasks successfully. Reminders that specify the situational context in which prospective memory cues will appear supply external information about the cue situation. When such reminders are available, individuals tend to rely on the provided external information to confirm cues, rather than allocating substantial attention resources to cue monitoring ([9]). Therefore, when reminders about the situation for prospective memory cues are given, children can successfully monitor those cues by mostly relying on reminders, thereby reducing the demand on their own attentional resources. This may compensate for their insufficient ability in cue monitoring.

Currently, many studies have found that reminders of contextual cues can significantly improve adults’ prospective memory performance ([2]; [15]), and this promoting effect can reduce their reliance on internal attention ([12]). However, currently only a small amount of research has focused on whether school-age children can effectively utilize the contextual information provided by reminders to improve prospective memory performance. For example, one study has examined how reminders for cue situation affect children’s prospective memory. [19] ([19]) instructed children in the reminder group to move the stimulus containing the prospective memory cues to the adjacent area of the response region in advance, thus facilitating task execution when the cues appeared. The results revealed that reminders enhanced children’s prospective memory performance. However, no cognitive offloading was observed in the reminder group, instead children’s task response speed even slowed down. This may be because the reminders placed by [19] ([19]) were positioned in a highly salient location of the task environment. Repeated exposure to reminder content may have progressively increased the activation level of the prospective memory representation, thereby enhancing children’s strategic monitoring for prospective memory cues. Moreover, the reminders employed by [19] ([19]) incorporated the specific content of the prospective memory task, thereby constituting reminders of the content itself. Consequently, whether reminders that solely specify the contextual circumstances in which prospective memory cues appear can significantly enhance children’s prospective memory performance remains to be empirically verified.

The facilitative effect of reminders on prospective memory may be influenced by attention loads. Under low attention load conditions, children possess sufficient available attention resources and can still allocate ample attention to cue monitoring of prospective memory. Consequently, the external information provided by reminders is unlikely to yield additional facilitation to children’s cue monitoring. However, under high attention load conditions, children have limited available attention resources and are unable to allocate sufficient attention toward monitoring prospective memory cues ([18]). Reminders that specify the situation in which prospective memory cues will appear provide external information about cue situations. Children can rely on this external information to confirm the presence of the cues, thereby compensating for their limited monitoring capacity. Therefore, the facilitative effect of reminders on prospective memory is more pronounced under high attention load conditions.

The distance between a reminder and the prospective memory cue may affect the facilitative impact of the reminder on prospective memory performance. When the reminder is presented in close temporal or spatial proximity to the prospective memory cue, participants may treat the reminder itself as an additional salient cue for the prospective memory tasks. Consequently, they are not required to engage in extensive cue monitoring, and they can rely primarily on the external information to determine the appropriate situation for executing prospective memory tasks ([22]). Thus, when reminders are situated near the prospective memory cue, children can successfully monitor the context where the prospective memory task is to be performed by relying on the external information supplied by the reminder, thereby significantly enhancing their prospective memory performance. However, when reminders are temporally or spatially distant from the prospective memory cue, it can only specify the approximate interval or region within which the cue is likely to appear, affording participants merely an expectation of the cue’s location. Under such conditions, individuals will sustain heightened cue monitoring specifically within the anticipated range where the prospective memory target may occur ([23]). Although reminders that are closer to prospective memory cues can significantly reduce their attentional expenditure on cue monitoring, the promoting effect of recent reminders on prospective memory may be influenced by attentional load. Under low load conditions, children may still retain sufficient attention to monitor prospective memory cues, so recent reminders may not be sufficient to significantly improve their prospective memory performance. However, under high attentional load conditions, children retain less attention for cue monitoring, and recent reminders may largely compensate for their lack of attention on cue monitoring and improve their prospective memory performance.

This study focused on the effects of reminders of the situations in which prospective memory cues appear on children’s prospective memory under different attention load conditions. We refer to reminders that occur relatively close to prospective memory cues as recent reminders and those that occur relatively far from prospective memory cues as distant reminders. Our specific objective was to examine whether only recent reminders could improve children’s prospective memory performance, and whether the promoting effect of recent reminders on children’s prospective memory was influenced by attentional load. Given that [19] ([19]) discovered the promoting effect of reminders on children’s prospective memory in the complex ongoing tasks, we hypothesized that only recent reminders under high attention load conditions could significantly improve children’s prospective memory accuracy. There was evidence to suggest that recent reminders could significantly reduce internal attention consumption, which reserved enough attention to process ongoing tasks ([12]). Therefore, we speculated that recent reminders would improve the accuracy of ongoing tasks under different attention load conditions. At present, there was no evidence indicating how reminders affect the response speed of prospective memory tasks and ongoing tasks, so we explored their relationship in an exploratory manner.

## 2. Method

### 2.1. Participants

Existing studies showed that the effect size of the facilitative effect of reminders on prospective memory ranged from moderate to large ([5]; [19]; [21]). Sample size was estimated using G*Power 3.1, with α = 0.05 and 1 − β = 0.80. When the effect size was large (*f* = 0.40), the calculated minimum sample size was 86. When the effect size was moderate (*f* = 0.25), the calculated minimum sample size was 211. We selected a sample size that falls between the two calculated minimum sample sizes mentioned above and ultimately recruited 192 primary school students aged 7–12 years, including 97 girls. Participants who forgot prospective memory tasks after the experiment or whose accuracy of ongoing tasks was lower than 0.5 were excluded, leaving 170 valid participants. Specifically, in the low attention load condition, 30 participants were in the recent reminder condition, 29 participants were in the distant reminder condition, and 29 participants in the control condition. Under the high attention load condition, 27 participants were in the recent reminder condition, 28 participants were in the distant reminder condition, and 27 participants were in the control condition. All participants were right-handed, had no color vision deficiency or mild color vision impairment, and had normal or corrected-to-normal vision. Before taking part in the experiment, the parents of all participants were asked to sign an informed consent form. We collected socio-demographic information, including age, gender, and years of education, and there was no difference in socio-demographic information among all groups (see Table 1).

### 2.2. Experimental Design

This study was an observational study and adopted a 3 (reminder conditions: recent reminder, distant reminder, control condition) × 2 (attention loads: low, high) between-subjects design. The accuracy and reaction time of prospective memory were primary variables, while the accuracy and reaction time of ongoing tasks were secondary variables.

### 2.3. Procedure

The experimental program was written and run with E-Prime 2.0 on one desktop computer, consisting of a practice phase and a formal experimental phase. At the start of the practice session, instructions for the ongoing task were provided. Each trial of ongoing tasks started with a “+” fixation point, which remained upon the screen for 500 milliseconds before disappearing. Then, Four-color blocks (the high attention load condition, see Figure 1) or two-color blocks (the low attention load condition, see Figure 2) were displayed in the center of the screen, persisting for 1500 milliseconds prior to vanishing. A blank screen was then presented as a buffer for 500 milliseconds. Finally, a colored word appeared, and participants were required to judge whether the color of the word had been presented in the previous color blocks. If it has occurred, participants should press the F key, or they should press the J key. Each word disappeared immediately upon response, with a maximum presentation duration of 4000 milliseconds. All words were selected from the commonly used characters in the first-grade primary school textbook (Volume 1) to ensure familiarity to all participants. The practice session comprises 20 ongoing task trials. If the accuracy of the ongoing task during the practice phase is higher than 0.8, the formal experimental phase can be entered, otherwise the practice will continue.

At the outset of the formal experiment, the instruction of prospective memory task appeared on the screen, informing participants to press the spacebar directly upon encountering either of the two words: flower or grass. After reading the instruction, participants needed to perform a 5 min reading filler task. This was mainly to demonstrate that prospective memory was a delayed memory task. Formal experiment comprised 60 trials, with 56 trials for the ongoing tasks and 4 trials for the prospective memory tasks. When prospective memory cues appeared, participants did not need to perform ongoing tasks and could directly perform prospective memory tasks. In the recent reminder condition, participants were informed that a red five-pointed star would appear in the same trial where the prospective memory task occurred. The five-pointed star would appear directly above the word. In the distant reminder condition, they were told that a red five-pointed star would appear 9 trials prior to prospective memory tasks. In order to prevent participants from forgetting the meaning of the five-pointed star, we would inform them that the five-pointed star was a reminder when the prospective memory instruction appeared and put a sticky note in front of the screen to write the meaning of the five-pointed star, thus avoiding them from forgetting. In the control condition, a red five-pointed star appeared in the first trial. Upon completion of the experiment, each participant was asked about the specific content of the prospective memory task to confirm whether participants still remembered it. The experimental procedure of this study was conducted only once.

## 3. Results

Prior to data analysis, we excluded trials with reaction time beyond 3 standard deviations. Then, we performed a normal distribution test (S-K test) on the remaining ongoing task data, and the data of the reaction time conformed to a normal distribution. However, the accuracy data did not conform to a normal distribution. Therefore, we used the log10 conversion method to convert the data of the ongoing task accuracy into a normal distribution. In addition, since prospective memory tasks only occurred a few times throughout the entire experiment, we did not perform a normal distribution test on the prospective memory performance. Our method for analyzing prospective memory performance was consistent with previous studies ([13]; [18]). This study used the general linear model in SPSS 20.0 statistical software to conduct a 3 (reminder conditions: recent reminder, distant reminder, control condition) × 2 (attention loads: low, high) analysis of variance on the accuracy and reaction time for both prospective memory tasks and ongoing tasks.

### 3.1. The Accuracy of Prospective Memory

An analysis of variance (ANOVA) was conducted on prospective memory accuracy. Results showed that the main effect of reminder conditions was significant, *F*(1, 164) = 6.20, *p* < 0.01, η*_p_*^2^ = 0.07. Post hoc multiple comparisons with Bonferroni correction indicated that the accuracy in the recent reminder condition was higher than that in the distant reminder condition (*p* = 0.02), and the accuracy in the recent reminder condition was also higher than that in the control condition (*p* = 0.001). Additionally, the main effect of attention loads was also significant, *F*(1, 164) = 21.65, *p* < 0.001, η*_p_*^2^ = 0.12. The accuracy of prospective memory in the high attention load condition was lower than that in the low attention load condition. However, the interaction effect between reminder conditions and attention loads was not significant, *F*(1, 164) = 0.27, *p* = 0.76 (See Figure 3).

### 3.2. The Response Time of Prospective Memory

An ANOVA was performed on the response time of prospective memory. The results showed that the main effect of attention loads was not significant, *F*(1, 164) = 3.63, *p* = 0.06. The main effect of reminder conditions was not significant, *F*(1, 164) = 0.70, *p* = 0.50. The interaction effect between reminder conditions and attention loads was not significant, *F*(1, 164) = 0.08, *p* = 0.93.

### 3.3. The Accuracy of Ongoing Tasks

An ANOVA was conducted on ongoing task accuracy. The results revealed a significant main effect of reminder conditions, *F*(1, 164) = 3.65, *p* = 0.01, η*_p_*^2^ = 0.04. Post hoc multiple comparisons with Bonferroni correction indicated that the accuracy rate in the recent reminder condition was higher than that in the distant reminder condition (*p* = 0.03), and the accuracy in the recent reminder condition was also higher than that in the control condition (*p* = 0.01). In addition, there was a significant main effect of attention loads, *F*(1, 164) = 125.24, *p* < 0.001, η*_p_*^2^ = 0.43. The accuracy under the high attention load condition was lower than that under the low attention load condition. However, the interaction effect between reminder conditions and attention loads was not significant, *F*(1, 164) = 0.68, *p* = 0.51 (see Figure 4).

### 3.4. The Response Time of Ongoing Tasks

An ANOVA was conducted on the response time of the ongoing task. The results revealed a significant main effect of attention loads, *F*(1, 164) = 6.68, *p* = 0.01, η*_p_*^2^ = 0.04, the reaction time under the high attention load condition was longer than that under the low attention load condition. However, the main effect of reminder conditions was not significant, *F*(1, 164) = 1.01, *p* = 0.37. The interaction between reminder conditions and attention loads was significant, *F*(1, 164) = 0.53, *p* = 0.59 (see Table 2).

## 4. Discussion

Reminders regarding the context in which prospective memory cues occur reduce individuals’ reliance on internal attention. Instead, they come to rely more on the external information provided by the reminders to determine when to perform prospective memory tasks ([12]). This may offset children’s inadequacies in attention monitoring, thereby enhancing their prospective memory performance. However, the facilitative effect of reminders on prospective memory may be influenced by the distance between the reminder and the cue, as well as the attention loads. This study aims to explore the influence of reminders at different distances from prospective memory cues on prospective memory under different attention loads. The results of this study found that the accuracy of prospective memory in children under low attention load conditions was higher than that under high attention load conditions, indicating that children’s prospective memory performance is easily influenced by attention load. According to the PAM theory, the preparatory attention processing of prospective memory relies heavily on attention, and preparatory attention processing is mainly used for monitoring cues. Therefore, the phenomenon that children’s prospective memory performance is easily affected by attention load indicates that it is difficult for children to effectively monitor cues when they have insufficient attention resources. In addition, the results revealed that the accuracy of prospective memory in the recent reminder condition was higher than that in both the distant reminder and the control condition, which was consistent with the results of a previous study ([19]). This suggests that only recent reminders can promote children’s prospective memory, which is consistent with our predictions. The failure of distant reminders to promote children’s prospective memory may be that there is still a considerable distance between recent reminders and prospective memory cues, which requires children to engage in cue monitoring for a relatively long time. Children have deficiencies in attention allocation and sustained attention ([25]; [28]), and their insufficient attention monitoring during intention maintenance may not be significantly compensated for under the condition of distant reminders. However, the recent reminder directly provides clear information about the location of the cue’s appearance. This enables children to successfully perform the prospective memory tasks by fully relying on external information ([12]). Recent reminders largely compensate for children’s attention inadequacies ([1]), thus significantly improving their prospective memory performance. However, the present study found that the accuracy of prospective memory in the recent reminder condition was higher than that in the control group under different attention loads. This shows that attention load does not influence the facilitative effect of recent reminders on children’s prospective memory. And this results conflict with our predictions. This might be because the prospective memory task used in this study was set at a comparatively high difficulty level, resulting in poor performance even under low attention load conditions. Meanwhile, the recent reminders we set were placed near the prospective memory cues, which substantially reduced the attention demands of prospective memory task. Therefore, children’s prospective memory performance remained influenced by the recent reminders even under the low attention load condition.

It is worth further exploration to examine how recent reminders promote prospective memory performance. Experimental studies on prospective memory tasks typically use a dual-task paradigm, in which the prospective memory task and the ongoing task need to be performed simultaneously ([3]). Thus, performance of the ongoing task can indirectly reflect the attention demands of the prospective memory task. The results of this study showed that the accuracy of the ongoing task in the recent reminder condition was higher than that in both the distant reminder condition and the control condition, which were consistent with the results of some previous studies ([5]). The results indicate that recent reminders reduce the attention demands of children’s prospective memory. Reminders provide information about the occurrence of prospective memory cues, enabling individuals to monitor these cues effectively while reducing attention demands ([5]). When performing ongoing tasks, prospective memory cues had not yet emerged. However, the results of the ongoing tasks in this study confirmed that children reduced their attention search for prospective memory cues in the presence of recent reminders. When reminders are present, individuals tend to rely more on the external information provided by the reminders, which causes them to reduce the internal attention’s monitoring of the cues ([9]). Thus, the finding in this study that attention demands are lower under the recent reminder condition reflects children have become more dependent on reminders. It remains to be examined whether the explicit information about the occurrence of prospective memory cues provided by recent reminders leads children in the reminder condition to respond promptly to prospective memory cues when they encounter the cues. Results regarding response speed of prospective memory showed no differences across different reminder conditions, indicating that even with reminders, children spent the same amount of time processing prospective memory tasks. When prospective memory cues emerge, individuals still need to go through processes such as cue identification, task switching, intention retrieval, and execution ([14]; [17]). Even under the recent reminder condition, children still engage in multiple subsequent cognitive processes. Consequently, when prospective memory cues appear, children do not exhibit the phenomenon of cognitive offloading. In conclusion, recent reminders prompt children to increase their reliance on external information to judge the context in which prospective memory cues appear, reducing their attention demands associated with cue monitoring during intention maintenance. Nevertheless, reminders fail to diminish the attention demands involved in retrieval of prospective memory intentions.

This study primarily examines the effects of reminders regarding the context in which prospective memory tasks occur on children’s prospective memory under varying attention load conditions. However, this study had several limitations. First, ongoing task performance was used as an indicator to reflect changes in attention, but it was only an indirect indicator. Eye-tracking technology, by contrast, can dynamically and in real time capture changes in an individual’s attention level as well as the specific processes involved in attention search, thereby functioning as a more direct and effective measure of attention ([26]). Second, children across different age groups exhibit varying abilities to utilize information provided by reminders ([6]). Although the present study focused exclusively on how children’s prospective memory is influenced by reminders, it did not explore whether such facilitative effects vary with age. Future research could build on this by employing eye-tracking technology to examine the impact of reminders on prospective memory among children of different ages. Future research can use eye tracking technology to investigate the relationship between the promoting effect of reminders on children’s prospective memory and attention, as well as whether the impact of reminders on children’s prospective memory changes with age. The results of this study indicate that only recent reminders can significantly improve children’s prospective memory performance, which has certain practical implications. When setting reminders for children in real life, we must ensure they are relatively close to the prospective memory cues. Only in this way can we increase the possibility of children successfully performing prospective memory tasks.

## 5. Conclusions

This present study examined the effects of reminders about the context in which prospective memory cues occur on children’s prospective memory ability under various levels of attention load conditions. The findings indicate that only recent reminders significantly improved children’s prospective memory accuracy, and this facilitative effect was not moderated by attention loads. In terms of theoretical significance, the results of this study not only clarify the scope of how reminders of contextual cues can promote children’s prospective memory performance, but also further examine the processing mechanism of reminders promoting children’s prospective memory. In practical implications, the results of this study suggest that when setting prompts for children in real life, we must ensure they are relatively close to the prospective memory cues in order to increase the likelihood of children successfully performing prospective memory tasks.

## Figures and Tables

**Figure 1 behavsci-15-01258-f001:**
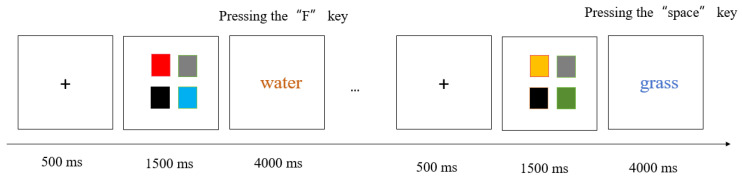
The experimental flowchart in the high load condition.

**Figure 2 behavsci-15-01258-f002:**
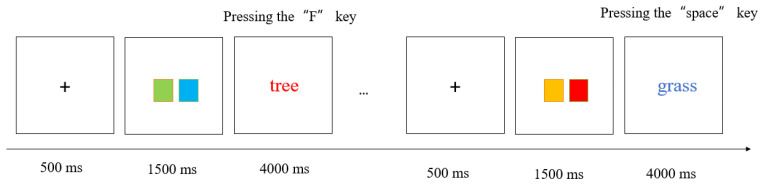
The experimental flowchart in the low load condition.

**Figure 3 behavsci-15-01258-f003:**
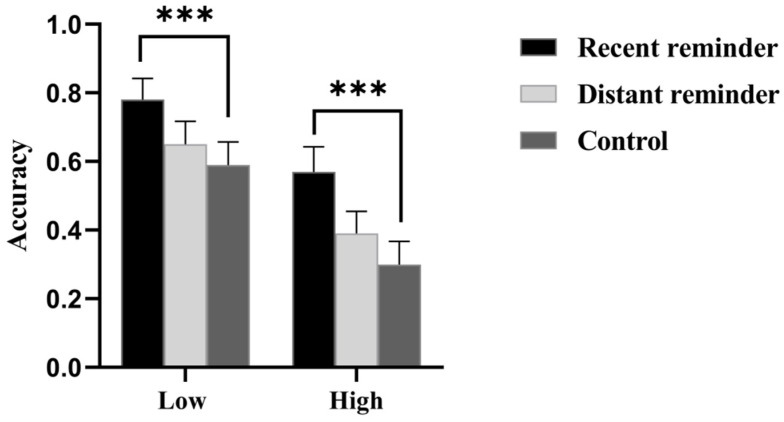
The prospective memory task accuracy. Notes: *** *p* < 0.001, Low = low attention load, High = high attention load.

**Figure 4 behavsci-15-01258-f004:**
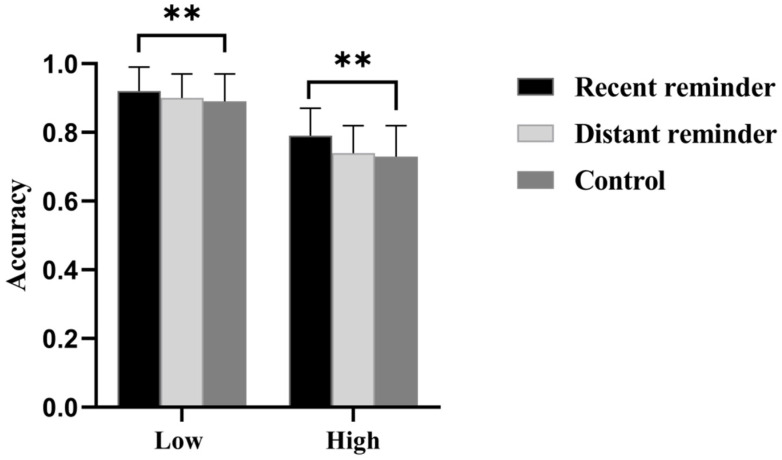
The Ongoing task accuracy. Notes: ** *p <* 0.01, Low = low attention load, High = high attention load.

**Table 1 behavsci-15-01258-t001:** The socio-demographic information in different groups.

		Age	The Proportion of Girls	Years of Education
Low	Recent	9.93 (1.34)	53%	4.03 (1.38)
	Distant	9.55 (1.92)	52%	3.55 (1.92)
	Control	9.31 (1.61)	52%	3.31 (1.61)
High	Recent	9.48 (1.74)	52%	3.48 (1.74)
	Distant	9.21 (1.75)	54%	3.21 (1.75)
	Control	9.59 (1.67)	52%	3.59 (1.67)

Note: Recent = recent reminder group, Distant = distant reminder group, Control = control group, Low = low attention load condition, High = high attention load condition, The Proportion of Girls= the proportion of girls in each group to the total number of people in that group.

**Table 2 behavsci-15-01258-t002:** The accuracy and reaction time of prospective memory task and ongoing task.

		ACC of PM	RT of PM (ms)	ACC of OT	RT of OT (ms)
Low	Recent	0.78 (0.34)	1080 (249)	0.92 (0.07)	1182 (227)
	Distant	0.65 (0.36)	1172 (232)	0.90 (0.07)	1256 (258)
	Control	0.59 (0.36)	1139 (248)	0.89 (0.08)	1217 (245)
High	Recent	0.57 (0.38)	1195 (421)	0.79 (0.08)	1331 (266)
	Distant	0.39 (0.34)	1240 (348)	0.74 (0.11)	1357 (243)
	Control	0.30 (0.35)	1232 (359)	0.73 (0.09)	1268 (273)

Note: Recent = recent reminder group, Distant = distant reminder group, Control = control group, Low = low attention load condition, High = high attention load condition, PM = prospective memory, OT = ongoing task, ACC = accuracy, RT = response time.

## Data Availability

The original contributions presented in this study are included in the article. Further inquiries can be directed to the corresponding author.

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
