# Peer review of "Recent Reminders with Word-Image Information Can Improve Children’s Prospective Memory Performance"

_behavsci, 2025, doi:10.3390/bs15091258_

Round 1
Reviewer 1 Report
Comments and Suggestions for Authors
The paper, "Only Recent Reminder Can Effectively Improve Children’s Prospective Memory Performance," presents an experimental study examining the influence of reminder conditions and cognitive load on prospective memory performance in children. The study showed that only recent reminders can improve children’s prospective memory. This research is significant as it involves children and investigates how the timing of reminders affects performance in a prospective memory task. However, the study lacks theoretical explanations and has some methodological shortcomings that need to be addressed. Detailed comments on the study are provided below.
MAJOR
ABSTRACT
- Please add information about the participants’ age mean and standard deviation.
INTRODUCTION
- Given that cue monitoring is a crucial aspect of the study, it would be beneficial to mention some theories related to bottom-up (automatic recognition) and top-down (strategic monitoring) processes in the introduction.
- It would be good to include some similar research involving adults, if available.
- On pg. 1, the authors reference existing research with children where the reminders focused on content rather than context. It would be useful to provide a brief description of these studies or explain how it is possible to separate content from context, considering that content reminders often also include the context in which a task must be performed.
- 3: There is a lack of explanation concerning the lower cognitive load and the distance between reminders and prospective memory cues. This aspect is also missing from the hypothesis.
- The researchers hypothesize that reminders will lead to faster reaction times in the ongoing task, but it is unclear where this assumption originates. Please clarify this point in the introduction.
METHODS
- There are several missing pieces of information regarding the determination of the sample size: 1) what effect size was used (d, f?) and why was an effect size of 0.4 chosen?; 2) for which analysis was the sample size estimated?; 3) was the sample size calculated per group or total? Furthermore, it is a bit confusing that despite the calculated required sample size, you refer to the study by Redshaw et al. (2018) when determining the number of participants.
- Please explain where the stars appear on the screen? Is it possible that seeing stars also requires monitoring the screen/environment? Why did you not use the auditory reminders? Also, how is showing the stars different from adding another prospective memory task in the recent reminder condition – did participants even have to read a word before performing the prospective memory task, after seeing the stars?
- Was a socio-demographic questionnaire used?
RESULTS
- At the beginning of the results, please write if a preliminary analysis have been conducted (outliers, normality of distributions, etc.).
- I recommend applying a correction for multiple comparisons in the post-hoc tests.
- Please add to pg. 4 (section Accuracy of Prospective Memory) and 5 (Accuracy of Ongoing Task) whether the difference between the distant reminder and control groups was statistically significant.
- The results in Table 1 are not consistent with the text on pg. 5, line 196.
DISCUSSION
- Pg, 6, lines 216 and 231: please add references of previous studies supporting your predictions/on which your predictions were based.
- Please refer to other studies when discussing better accuracy in the ongoing task in the recent reminder group compared to the control group? I would expect opposite results due to the limited time and cognitive resources allocated to completing the prospective memory task.
- 6, lines 247-248: provide an explanation in the methods section regarding how prospective memory cues did not emerge while performing the ongoing task.
- Practical implications are missing.
MINOR
INTRODUCTION
- 2, line 48, change “arent’t” to “are not”.
- 2, line 65 and 69: when mentioning Redshaw et al., please always write the full reference
RESULTS
- 5, line 196: I suggest using the term “reaction time” instead of “reaction speed”.
REFERENCES
- Please add doi each reference.
Author Response
Reviewer 1
The paper, "Only Recent Reminder Can Effectively Improve Children’s Prospective Memory Performance," presents an experimental study examining the influence of reminder conditions and cognitive load on prospective memory performance in children. The study showed that only recent reminders can improve children’s prospective memory. This research is significant as it involves children and investigates how the timing of reminders affects performance in a prospective memory task. However, the study lacks theoretical explanations and has some methodological shortcomings that need to be addressed. Detailed comments on the study are provided below.
ABSTRACT
- Please add information about the participants’ age mean and standard deviation.
Response: Thanks for your suggestion! We have added the mean age and standard deviation of the participants in the Abstract section. (Line 11)
INTRODUCTION
- Given that cue monitoring is a crucial aspect of the study, it would be beneficial to mention some theories related to bottom-up (automatic recognition) and top-down (strategic monitoring) processes in the introduction.
Response: Thanks for your suggestion! We have added descriptions of the theories of preparatory attentional processing and memory processing in the Introduction, and pointed out that top-down preparatory attentional processing mainly refers to the prospective component, while bottom-up memory processing is primarily involved in the retrospective component. (Line 27-36)
- It would be good to include some similar research involving adults, if available.
Response: Thanks for your suggestion! We have included some similar research involving adults.(Line 74-46)
- On pg. 1, the authors reference existing research with children where the reminders focused on content rather than context. It would be useful to provide a brief description of these studies or explain how it is possible to separate content from context, considering that content reminders often also include the context in which a task must be performed.
Response: Thanks for your suggestion! It is true that many content reminders often also include the context in which a task must be performed. However, if a reminder only presents warning information (e.g., setting an alarm) and does not display the specific content of prospective memory, such a reminder merely indicates when the prospective memory cue will appear, without informing the individual of the content of the prospective memory. This makes it possible to separate the content information from the information about when the cue appears. We have added descriptions of the differences between content reminders and contextual reminders in the Introduction. (Line 53-57)
- There is a lack of explanation concerning the lower cognitive load and the distance between reminders and prospective memory cues. This aspect is also missing from the hypothesis.
Response: Thanks for your suggestion! We have added an explanation concerning the lower cognitive load and the distance between reminders and prospective memory cues. (Line 122-130)
- The researchers hypothesize that reminders will lead to faster reaction times in the ongoing task, but it is unclear where this assumption originates. Please clarify this point in the introduction.
Response: Thanks for your suggestion! There is insufficient evidence to support the facilitation of task response speed, so we have removed this hypothesis.
METHODS
- There are several missing pieces of information regarding the determination of the sample size: 1) what effect size was used (d, f?) and why was an effect size of 0.4 chosen?; 2) for which analysis was the sample size estimated?; 3) was the sample size calculated per group or total? Furthermore, it is a bit confusing that despite the calculated required sample size, you refer to the study by Redshaw et al. (2018) when determining the number of participants.
Response: Thanks for your suggestion! (1) The effect size we selected is f=0.4. (2) Several studies we referenced found that the effect size of reminders' facilitative effect on prospective memory is relatively large (Chen et al., 2017; Redshaw et al., 2018) or moderate (Ryder et al., 2022). Although a large effect size was selected in this study, we referred to the sample sizes of previous studies and ultimately recruited 192 participants, among whom 170 were valid participants. (using G*Power for analysis, when f = 0.4, the minimum sample size is 86, when f = 0.25,the minimum sample size is 211) . The sample size of this study falls between the range corresponding to a moderate effect size and a large effect size, which is consistent with the effect sizes reported in previous studies. (3) The sample size in this study refers to the total number of participants. We have added relevant explanations in the Participants section. (4) We estimated the required sample size using G*Power software and considered the sample sizes reported in previous studies. This approach was taken to ensure that the sample size selected for this study is well-justified on multiple grounds.
References
Chen, Y., Lian, R., Yang, L., Liu, J., & Meng, Y. (2017). Working memory load and reminder effect on event-based prospective memory of high-and low-achieving students in math. Journal of learning disabilities, 50 (5), 602-608.
Redshaw, J., Vandersee, J., Bulley, A., & Gilbert, S. J. (2018). Development of children's use of external reminders for hard‐to‐remember intentions. Child Development, 89 (6), 2099-2108.
Ryder, N., Kvavilashvili, L., & Ford, R. (2022). Effects of incidental reminders on prospective memory in children. Developmental Psychology, 58 (5), 890-903.
- Please explain where the stars appear on the screen? Is it possible that seeing stars also requires monitoring the screen/environment? Why did you not use the auditory reminders? Also, how is showing the stars different from adding another prospective memory task in the recent reminder condition – did participants even have to read a word before performing the prospective memory task, after seeing the stars?
Response: Thanks for your suggestion! The stars appear directly above the adjacent word, and we have added relevant descriptions in the text. This pentagram is easy to notice and does not consume excessive additional attention. Auditory reminders are also effective; however, most previous studies have used visual reminders, so we have also adopted visual graphics. Participants see the pentagram simultaneously with the presentation of the word. The pentagram is a salient external cue, which serves to inform participants that the currently presented stimulus is the prospective memory cue. In this way, prospective memory cues can be effectively retrieved without the need for attentional monitoring at all. (Line 199-200)
- Was a socio-demographic questionnaire used?
Response: Thanks for your suggestion! We collected basic sociodemographic information of the participants, including their age, gender, and educational level. We have added relevant descriptions in the text. There were no differences in these sociodemographic characteristics across the different groups. (Line 156-159)
RESULTS
- At the beginning of the results, please write if a preliminary analysis have been conducted (outliers, normality of distributions, etc.).
Response: Thanks for your suggestion! We have added descriptions of the relevant preliminary analyses at the beginning of the Results section. Before data analysis, we excluded data from trials where the reaction time fell outside 3 standard deviations. Subsequently, we conducted a normality test (Shapiro-Wilk test) on the remaining ongoing task data: the reaction time data conformed to a normal distribution, while the accuracy data did not. Additionally, since the prospective memory task only appeared a few times throughout the entire process, we did not conduct a normality test on the performance of the prospective memory task. The method used in this study to analyze prospective memory performance is consistent with that employed in other previous studies (Haines et al., 2020; Peper et al., 2023). (Line 210-221)
- I recommend applying a correction for multiple comparisons in the post-hoc tests.
Response: Thanks for your suggestion! We applied the Bonferroni correction for multiple comparisons in the post-hoc tests. (Line 226, Line 246)
- Please add to pg. 4 (section Accuracy of Prospective Memory) and 5 (Accuracy of Ongoing Task) whether the difference between the distant reminder and control groups was statistically significant.
Response: Thanks for your suggestion! We have added relevant descriptions in the corresponding section to explain whether the difference between the distant reminder group and the control group was statistically significant. (Line 228-229, Line 259-261)
- The results in Table 1 are not consistent with the text on pg. 5, line 196.
Response: Thanks for your suggestion! We have corrected this error in the text. (Line 259-260)
DISCUSSION
- Pg, 6, lines 216 and 231: please add references of previous studies supporting your predictions/on which your predictions were based.
Response: Thanks for your suggestion! We have added two references to support our predictions. (Line 300-303)
- Please refer to other studies when discussing better accuracy in the ongoing task in the recent reminder group compared to the control group? I would expect opposite results due to the limited time and cognitive resources allocated to completing the prospective memory task.
Response: Thanks for your suggestion! We have added the corresponding references. Since reminders reduce attentional resources allocated to cue monitoring, this allows individuals to reserve more attention for the processing of the ongoing task, thereby improving the performance of the ongoing task. (Line 320-322)
- Pg, 6, lines 247-248: provide an explanation in the methods section regarding how prospective memory cues did not emerge while performing the ongoing task.
Response: Thanks for your suggestion! We have added the corresponding explanations in the Procedure section. (196-197)
- Practical implications are missing.
Response: Thanks for your suggestion! We have added content regarding the practical implications of this study's results in the final paragraph of the Discussion section. (Line 365-368)
MINOR
INTRODUCTION
18, line 48, change “arent’t” to “are not”.
Response: Thanks for your suggestion! We have corrected this error. (Line 54)
19, line 65 and 69: when mentioning Redshaw et al., please always write the full reference
Response: Thanks for your suggestion! We have corrected this error. (Line 86, Line 90)
RESULTS
20, line 196: I suggest using the term “reaction time” instead of “reaction speed”.
Response: Thanks for your suggestion! We have used the term "reaction time" instead of "reaction speed". (Line 260)
REFERENCES
- Please add doi each reference.
Response: Thanks for your suggestion! We have added DOIs after all the references. (Line 393)
Reviewer 2 Report
Comments and Suggestions for Authors
This study was aimed at investigating prospective memory (PM) functioning in primary school children (age: 7-12 years) with a specific focus on the effect of reminders. The results show that, in this population, PM performance may be related to specific characteristics of the reminders indicating a facilitation effect for recent PM-reminders. The experimental design and methods are sound to investigate the hypotheses of the study and findings are well discussed.
This study investigated an important topic in the field and it may be of interest for the readers of the Journal.
Here below I report, point-by-point, some suggestions that could be taken into account in a revision.
- The experimental procedure and stimuli should be better described. It could be useful to include in the manuscript a figure illustrating procedure and stimuli;
- The results presentation should be improved (also null effects should be reported). At this regard, I would suggest to insert a paragraph in order to better describe statistical models and factors included in the analyses;
- PM performance appears to be significantly affected by the attentional load. Although the results indicate that this effect is not specific to the recent reminders condition it should be more deeply discussed.
Author Response
Reviewer 2
This study was aimed at investigating prospective memory (PM) functioning in primary school children (age: 7-12 years) with a specific focus on the effect of reminders. The results show that, in this population, PM performance may be related to specific characteristics of the reminders indicating a facilitation effect for recent PM-reminders. The experimental design and methods are sound to investigate the hypotheses of the study and findings are well discussed.
This study investigated an important topic in the field and it may be of interest for the readers of the Journal.
Here below I report, point-by-point, some suggestions that could be taken into account in a revision.
- The experimental procedure and stimuli should be better described. It could be useful to include in the manuscript a figure illustrating procedure and stimuli;
Response: Thanks for your suggestion! We have added experimental flowcharts for both the low-load condition and the high-load condition in the text. (Line 185-188)
- The results presentation should be improved (also null effects should be reported). At this regard, I would suggest to insert a paragraph in order to better describe statistical models and factors included in the analyses;
Response: Thanks for your suggestion! (1) We have supplemented the results of the null effects. (2) We have added a paragraph at the beginning of the Results section to describe the statistical models and factors used in the data analysis of this study. (Line 210-222)
- PM performance appears to be significantly affected by the attentional load. Although the results indicate that this effect is not specific to the recent reminders condition it should be more deeply discussed.
Response: Thanks for your suggestion! We have added a discussion on the susceptibility of prospective memory performance to attentional load in the Discussion section. (Line 270-284)
Reviewer 3 Report
Comments and Suggestions for Authors
This manuscript provides interesting information in the field of neuroscience for children aged 7 to 12 years. This can help establish educational and cognitive performance strategies in school, social, and family settings.
This manuscript presents some shortcomings that need to be addressed with various contributions, clarifications, and improvements, which I outline below:
-At the end of the introduction, the authors should establish a general objective and specific objectives. They should then state a hypothesis.
-They should establish a structured narrative abstract. The authors should present the abstract in the following order: very brief introduction, general objective, design, participants and location, instruments, procedure (referring to the research ethics committee), results (with the most relevant numerical and statistical data related to the specific objectives initially stated), and a very brief discussion and conclusions.
-In the introduction, the authors should provide prevalence data related to the central topic they are discussing and will research (at the global, national, and local levels, if available). In addition, they must explain the controversy and the most recent studies that support the initial hypothesis.
-The authors must explain the type of design: observational study, repeated measures study, etc. The authors must clarify this section.
-In the design section, the authors must explain the variables to be measured as primary and secondary variables.
-The authors must add an instrument section, where they describe the type of tests used, the original authors and the validation authors of the instruments or tests, describe the test, method of administration, and administration time, and provide normative data if available.
-The authors must describe the procedure in detail, phased by phase, so that any other researcher can replicate the study (Phase 1, Phase 2, Phase 3, etc.). They must refer to the approval of the Research Ethics Committee (report number and approval date).
-In the abstract and results section, authors should present statistical data using APA style (it is important that authors review this style, although they closely resemble it in their writing and presentation).
In APA style, means (M) and standard deviations (SD) are presented concisely and clearly, usually within the text or in tables. For presentation in the text, the mean is used with one more decimal place than the original data, and the standard deviation is used with the same number of decimal places as the mean. The abbreviations "M" for mean and "SD" for standard deviation are used.
Presentation in the text:
- Mean:
Written with a lowercase letter "M" followed by an equal sign and the mean value. For example, "M = 10.5."
- Standard Deviation:
Written with a capital letter "SD" followed by an equal sign and the standard deviation value. For example, "SD = 2.1."
- Example:
"Participants obtained a mean score of 72.5 (SD = 8.2) on the test."
Presentation in tables:
- Tables should have a number (in bold) and a brief title in italics.
- Columns should be clear and concise, including the mean and standard deviation, usually in the same row or column.
- It is recommended to avoid vertical lines in tables and use only horizontal lines necessary for clarity.
Additional Considerations:
- If the data distribution is not normal, it is recommended to report other measures of dispersion, such as quartiles or percentiles.
- For p-values less than .001, p < .001 is reported.
- It is recommended to be consistent with the number of decimal places throughout the document.
- Tables should be self-explanatory enough for the reader to understand the analyses without having to consult the main text.
-The results should be presented in order of the objectives, to address them and contrast the hypothesis. They should present a table with the basic sociodemographic data: gender, age, educational level. Observe the contrast of means for gender to control for this variable, since the authors have a considerable sample size.
-The discussion begins with the general objective. The discussion should be developed and structured in order of the specific objectives they propose. The authors should present, contrast, interpret, and discuss the sociodemographic data. They should use more up-to-date references, primarily from the last 5 years, especially in the discussion.
At the end of the discussion, they should clearly state the limitations (preferably numbered), future directions, and the impact and relevance the results may have at the academic, clinical, and other levels.
- The conclusions should address the new specific objectives initially stated. They may also conclude with a future proposal containing the conclusions obtained regarding the use and implementation of protocols, approach strategies, etc.
-The bibliographic references should be in APA style, but must include the DOIs in the following format, for example: https://doi.org/10.11604/pamj.2018.30.24.15230.K
-It is recommended that authors use a proofreading service to review their scientific narratives.
Comments on the Quality of English Language-It is recommended that authors use a proofreading service to review their scientific narratives.
Author Response
Reviewer 3
This manuscript provides interesting information in the field of neuroscience for children aged 7 to 12 years. This can help establish educational and cognitive performance strategies in school, social, and family settings.
This manuscript presents some shortcomings that need to be addressed with various contributions, clarifications, and improvements, which I outline below:
- At the end of the introduction, the authors should establish a general objective and specific objectives. They should then state a hypothesis.
Response: Thanks for your suggestion! We established a general objective and specific objectives in the final paragraph of the Introduction, and stated a hypothesis based on these objectives at the end. (Line 131-140)
- They should establish a structured narrative abstract. The authors should present the abstract in the following order: very brief introduction, general objective, design, participants and location, instruments, procedure (referring to the research ethics committee), results (with the most relevant numerical and statistical data related to the specific objectives initially stated), and a very brief discussion and conclusions.
Response: Thanks for your suggestion! We have rewritten the abstract section of this study using the aforementioned structure. (Line 8-20)
- In the introduction, the authors should provide prevalence data related to the central topic they are discussing and will research (at the global, national, and local levels, if available). In addition, they must explain the controversy and the most recent studies that support the initial hypothesis.
Response: Thanks for your suggestion! We have added descriptions of the current status of prospective memory ability in children globally and in China locally. In addition, currently, only one study has focused on the facilitative effect of contextual reminders on children’s prospective memory. We have added relevant descriptions in the Introduction to support our hypothesis. (Line 41-45)
- The authors must explain the type of design: observational study, repeated measures study, etc. The authors must clarify this section.
Response: Thanks for your suggestion! This study was an observational study. We have added this description in the Experimental Design section. (Line 161)
- In the design section, the authors must explain the variables to be measured as primary and secondary variables.
Response: Thanks for your suggestion! The accuracy rate and reaction time of prospective memory are the primary variables, while the accuracy rate and reaction time of the ongoing task are the secondary variables. We have added this description in the Experimental Design section. (Line 163-164)
- The authors must add an instrument section, where they describe the type of tests used, the original authors and the validation authors of the instruments or tests, describe the test, method of administration, and administration time, and provide normative data if available.
Response: Thanks for your suggestion! The experimental program was written and run with E-Prime 2.0 on one desktop computer. Since no other instruments or equipment were involved except for one computer and E-Prime software, we ultimately added the information about the instruments to the first sentence of the Procedure section instead of creating a separate "Instrument Section". (Line 170-171)
- The authors must describe the procedure in detail, phased by phase, so that any other researcher can replicate the study (Phase 1, Phase 2, Phase 3, etc.). They must refer to the approval of the Research Ethics Committee (report number and approval date).
Response: Thanks for your suggestion! The experimental procedure of this study was conducted only once, so multi-stage measurement results cannot be reported. Our single phase has obtained approval from the Institutional Review Board of Henan Provincial Key Laboratory of Psychology and Behavior (protocol code: 20250509001, date of approval: 2025-02-24). (Line 208, Line 385-387)
- In the abstract and results section, authors should present statistical data using APA style (it is important that authors review this style, although they closely resemble it in their writing and presentation). In APA style, means (M) and standard deviations (SD) are presented concisely and clearly, usually within the text or in tables. For presentation in the text, the mean is used with one more decimal place than the original data, and the standard deviation is used with the same number of decimal places as the mean. The abbreviations "M" for mean and "SD" for standard deviation are used.
Presentation in the text:
Mean: Written with a lowercase letter "M" followed by an equal sign and the mean value. For example, "M = 10.5."
Standard Deviation: Written with a capital letter "SD" followed by an equal sign and the standard deviation value. For example, "SD = 2.1."
Example: "Participants obtained a mean score of 72.5 (SD = 8.2) on the test."
Presentation in tables:
Tables should have a number (in bold) and a brief title in italics.
Columns should be clear and concise, including the mean and standard deviation, usually in the same row or column.
It is recommended to avoid vertical lines in tables and use only horizontal lines necessary for clarity.
Additional Considerations:
If the data distribution is not normal, it is recommended to report other measures of dispersion, such as quartiles or percentiles.
For p-values less than .001, p < .001 is reported.
It is recommended to be consistent with the number of decimal places throughout the document.
Tables should be self-explanatory enough for the reader to understand the analyses without having to consult the main text.
Response: Thanks for your suggestion! We have revised the format of this manuscript in accordance with the APA style. (1) Data are reported with two decimal places consistently (except when the p-value is less than 0.001). (2) We have added detailed explanations under Table 1 and Table 2, allowing readers to understand the specific content therein without referring to other parts of the results. (3) For data that did not conform to a normal distribution, we performed a log10 transformation to make the transformed data conform to a normal distribution. (4) The format of the table was revised by the editorial staff during the initial review. Although we agree that your suggestion is more in line with the standard format specified by APA, we are unable to make further modifications to it
- The results should be presented in order of the objectives, to address them and contrast the hypothesis. They should present a table with the basic sociodemographic data: gender, age, educational level. Observe the contrast of means for gender to control for this variable, since the authors have a considerable sample size.
Response: Thanks for your suggestion! We have added Table 1 to present the basic sociodemographic data: gender, age, and educational level.
- The discussion begins with the general objective. The discussion should be developed and structured in order of the specific objectives they propose. The authors should present, contrast, interpret, and discuss the sociodemographic data. They should use more up-to-date references, primarily from the last 5 years, especially in the discussion.
Response: Thanks for your suggestion! (1) We have reorganized the content of the Discussion section. Currently, the Discussion section includes the following aspects: research background, research results, comparison with previous studies, mechanism analysis, as well as research significance and limitations. (2) We have added the latest literature in the Discussion section, among which the proportion of literature published in the recent five years exceeds half. (3) We have also added the analysis and discussion of sociodemographic data in the Discussion section. (Line 288-291)
- At the end of the discussion, they should clearly state the limitations (preferably numbered), future directions, and the impact and relevance the results may have at the academic, clinical, and other levels.
Response: Thanks for your suggestion! We have added the limitations of the study in the final paragraph of the Discussion section, which are marked with serial numbers. Additionally, we have included the directions for future research and the practical significance of this study. (350-368)
- The conclusions should address the new specific objectives initially stated. They may also conclude with a future proposal containing the conclusions obtained regarding the use and implementation of protocols, approach strategies, etc.
Response: Thanks for your suggestion! We have summarized the specific conclusions of this study in the Conclusion section, and these conclusions are consistent with the specific hypotheses proposed in the Introduction. Additionally, at the end of the Conclusion, we have summarized the academic significance of this study and put forward recommendations closely related to real life based on these conclusions. (Line 371-380)
-The bibliographic references should be in APA style, but must include the DOIs in the following format, for example: https://doi.org/10.11604/pamj.2018.30.24.15230.K
Response: Thanks for your suggestion! We have added DOIs after all the references. (Line 394-456)
-It is recommended that authors use a proofreading service to review their scientific narratives.
Response: Thanks for your suggestion! We have asked an editor from a specialized language polishing agency to revise the language issues for us, and we have also requested a full professor of psychology to revise the psychology-related professional terminology involved in this research.
Round 2
Reviewer 1 Report
Comments and Suggestions for Authors
The authors of the text "Only Recent Reminder Can Effectively Improve Children’s Prospective Memory Performance" have significantly improved the paper. However, there are certain issues that need to be addressed before the manuscript can be accepted.
INTRODUCTION
- In the introduction, it is mentioned that the retrospective component is primarily a bottom-up process. However, I am uncertain if this conclusion is supported by the study you referenced (Guo et al., 2024). Please verify this statement.
- Currently, the introduction includes a hypothesis only for the dependent variable of prospective memory accuracy. However, the researchers also examine additional dependent variables, such as response time in the prospective memory task, accuracy, and response time in the ongoing task. It is necessary to formulate hypotheses for each of these dependent variables, or alternatively, indicate that these relationships will be investigated in an exploratory manner, if there is insufficient data in the literature to form specific hypotheses.
METHODS
- Table 1, Figure 1, and Figure 2 should be referenced in the text.
- The explanation you provided regarding sample size determination should be included in the text. It should detail how 'f' was determined by referencing previous studies and clarify why a larger sample size was utilized.
DISCUSSION
- I suggest removing the following sentence from the discussion (pg. 8): “We found no differences in socio-demographic information, including age, gender, and education years, among different groups, indicating that the promoting effect of reminders on children's prospective memory is not caused by the interference of socio-demographic variables.”
Author Response
The authors of the text "Only Recent Reminder Can Effectively Improve Children’s Prospective Memory Performance" have significantly improved the paper. However, there are certain issues that need to be addressed before the manuscript can be accepted.
INTRODUCTION
- In the introduction, it is mentioned that the retrospective component is primarily a bottom-up process. However, I am uncertain if this conclusion is supported by the study you referenced (Guo et al., 2024). Please verify this statement.
Response: Thanks for your suggestion! The view that the retrospective component is primarily a bottom-up process is a key tenet of the preparatory attentional processing and memory processing theory. However, we have not yet found any direct studies that can verify this view. Therefore, we ultimately removed this description.
- Currently, the introduction includes a hypothesis only for the dependent variable of prospective memory accuracy. However, the researchers also examine additional dependent variables, such as response time in the prospective memory task, accuracy, and response time in the ongoing task. It is necessary to formulate hypotheses for each of these dependent variables, or alternatively, indicate that these relationships will be investigated in an exploratory manner, if there is insufficient data in the literature to form specific hypotheses.
Response: Thanks for your suggestion! We have added predictions for the accuracy of ongoing tasks based on previous research perspectives. However, there was no evidence indicating how reminders affect the response speed of prospective memory tasks and ongoing tasks, so we explored their relationship in an exploratory manner. We have added relevant descriptions in the last paragraph of the Introduction. (Line 140-146)
METHODS
- Table 1, Figure 1, and Figure 2 should be referenced in the text.
Response: Thanks for your suggestion! We have cited Table 1, Figure 1 and Figure 2 in the text. (Line 167, 183)
- The explanation you provided regarding sample size determination should be included in the text. It should detail how 'f' was determined by referencing previous studies and clarify why a larger sample size was utilized.
Response: Thanks for your suggestion! Existing studies showed that the effect size of the facilitative effect of reminders on prospective memory ranged from moderate to large (Chen et al., 2017; Redshaw et al., 2018; Ryder et al., 2022). Sample size was estimated using G*Power software, with α = 0.05 and 1-β = 0.80. When the effect size was large (f = 0.40), the calculated minimum sample size was 86. When the effect size was moderate (f = 0.25), the calculated minimum sample size was 211. We selected a sample size that falls between the two calculated minimum sample sizes mentioned above. We have added relevant descriptions in the Participants section. (Line 149-154)
DISCUSSION
- I suggest removing the following sentence from the discussion (pg. 8): “We found no differences in socio-demographic information, including age, gender, and education years, among different groups, indicating that the promoting effect of reminders on children's prospective memory is not caused by the interference of socio-demographic variables.”
Response: Thanks for your suggestion! We have removed the sentence.
Reviewer 3 Report
Comments and Suggestions for Authors
The authors have made the relevant clarifications and corrections in their most approximate form.
Author Response
The authors have made the relevant clarifications and corrections in their most approximate form.
Response: Thank you for your approval of our revised manuscript!